# Silybin-Induced Apoptosis Occurs in Parallel to the Increase of Ceramides Synthesis and miRNAs Secretion in Human Hepatocarcinoma Cells

**DOI:** 10.3390/ijms20092190

**Published:** 2019-05-03

**Authors:** Silvia Zappavigna, Daniela Vanacore, Stefania Lama, Nicoletta Potenza, Aniello Russo, Pasquale Ferranti, Marcello Dallio, Alessandro Federico, Carmelina Loguercio, Pasquale Sperlongano, Michele Caraglia, Paola Stiuso

**Affiliations:** 1Department of Precision Medicine, University of Campania “Luigi Vanvitelli”, Via De Crecchio 7, 80138 Naples, Italy; silvia.zappavigna@unicampania.it (S.Z.); daniela.vanacore@unicampania.it (D.V.); stefania.lama@unicampania.it (S.L.); marcello.dallio@gmail.com (M.D.); alessandro.federico@unicampania.it (A.F.); carmelina.loguercio@unicampania.it (C.L.); michele.caraglia@unicampania.it (M.C.); 2Department of Environmental, Biological and Pharmaceutical Sciences and Technologies, University of Campania “L. Vanvitelli”, viale Lincoln, 81100 Caserta, Italy; nicoletta.potenza@unicampania.it (N.P.); aniello.russo@unicampania.it (A.R.); 3Department of Agricultural Sciences, University of Naples Federico II, Via Università, 100, 80055 Portici, NA, Italy; pasquale.ferranti@unina.it; 4Department of Translational Medical Sciences, University of Campania “Luigi Vanvitelli”, Via De Crecchio 7, 80138 Naples, Italy; pasquale.sperlongano@unicampania.it

**Keywords:** apoptosis, ceramides, hepatocarcinoma, miRNAs, PTEN, sorafenib, silybin

## Abstract

Silybin is a flavonolignan extracted from *Silybum marianum* (milk thistle) with hepatoprotective, antioxidant, and anti-inflammatory activity. Several studies have shown that silybin is highly effective to prevent and treat different types of cancer and that its antitumor mechanisms involve the arrest of the cell cycle and/or apoptosis. An MTT assay was performed to study cell viability, lipid peroxidation, extracellular NO production, and scavenger enzyme activity were studied by Thiobarbituric Acid-Reactive Species (TBARS) assay, NO assay, and MnSOD assay, respectively. Cell cycle and apoptosis analysis were performed by FACS. miRNA profiling were evaluated by real time PCR. In this study, we demonstrated that Silybin induced growth inhibition blocking the Hepg2 cells in G1 phase of cell cycle and activating the process of programmed cell death. Moreover, the antiproliferative effects of silybin were paralleled by a strong increase of the number of ceramides involved in the modulation of miRNA secretion. In particular, after treatment with silybin, miR223-3p and miR16-5p were upregulated, while miR-92-3p was downregulated (*p* < 0.05). In conclusion, our results suggest that silybin-Induced apoptosis occurs in parallel to the increase of ceramides synthesis and miRNAs secretion in HepG2 cells.

## 1. Introduction

Silybin (sil), a polyphenolic flavonoid, is the main component of silymarin with antioxidant and free radical scavenger activity, used as dietary supplement in the treatment of alcoholic liver disease [1]. We have recently reported that a chronic treatment with Realsil—a food supplement in which silybin is conjugated with phosphatidylcholineand vitamin E —significantly decreases liver damage plasma marker levels (AST, ALT, and γGT) and improves liver histology in ~50% of patients with NAFLD and NASH [2]. Moreover, we demonstrated that Realsil treatment restored the normal serum lipidomic profile in NAFDL patients by decreasing the risk to develop NASH-related hepatocellular carcinoma (HCC) [3,4,5,6]. Hepatocellular carcinoma (HCC) is the most common form of hepatic cancer with a poor prognosis in the advanced stage. Liver cirrhosis is the main condition that predisposes to HCC and is associated with it in ~90% of cases. Chronic infection with major hepatitis viruses (HBV and HCV) is the major cause of cirrhosis and therefore, indirectly, also of HCC [7,8]. The main treatment options for patients with HCC include tumor ablation, surgery, and transplantation, but unfortunately, these are not effective in patients with an advanced stage of disease [9,10].

On these bases, to develop a therapeutic strategy that is effective in this group of patients is of fundamental importance [11]. Recently, significant progress has been made in the understanding of molecular mechanisms underlying hepatocarcinogenesis by leading to the development of target-based drugs [12]. The Sorafenib HCC Assessment Randomized Protocol (SHARP) represents the first international phase 3, randomized trial that demonstrated the efficacy of sorafenib (Bayer 43-9006; Nexavar, Bayer AG Inc., Leverkusen, Germany) in prolonging median survival in HCC patients [10]. Sorafenib is a multikinase inhibitor that acts on Raf and VEGF pathways by inhibiting proliferation and angiogenesis in several tumor models [13,14]. Currently, Sorafenib is approved by the US Food and Drug Administration for the treatment of patients with advanced HCC [14].

MicroRNAs (miRNA) are small noncoding RNA molecules of 21–22 nucleotides in length that post-transcriptionally regulate gene expression and play a key role in carcinogenesis by acting as oncogenes or tumor suppressor [15]. We have recently demonstrated that sorafenib increases the release in vitro of miR423-5p from HCC cell lines and enhances the miR423-5p secretion in the serum of patients affected by HCC. The in vitro effects were also paralleled by a change of the lipidomic profile of HCC cell lines with an increase of ceramides. The latter are involved in the regulation of miRNA secretion from normal and tumor cells [16]. Several studies have shown that silybin is highly effective to prevent and treat different types of cancer and that its antitumor mechanisms involve the arrest of the cell cycle and/or apoptosis [17], but a better understanding of the molecular mechanisms underlying these effects is needed. In this study we have explored the in vitro silybin proapoptotic ability and evaluated the effects of silybin in combination with sorafenib on HCC cell growth.

## 2. Results

### 2.1. Silybin Induced Cell Cycle Block and Apoptosis in HCC Cell Lines by Nitric Oxide Production

We evaluated the cytotoxic effects of different concentrations of silybin (Figure 1) on HepG2 cell growth; growth inhibition was time- and dose-dependent. In particular, the results obtained by the MTT assay showed that silybin induced 50% growth inhibition at 68 μM (IC50) after 72 h of treatment.

The antioxidant activity of silybin was studied evaluating lipid peroxidation by Thiobarbituric Acid-Reactive Species (TBARS) assay, extracellular NO production, and the scavenger enzyme activity. Silybin treatment induced a significant decrease of TBARS values compared to control cells. On the contrary, both extracellular nitric oxide (NO) production and MnSOD activity increased when HepG2 cells were treated for 72 h with silybin compared to untreated cells (Figure 2).

Recent studies reported that nitric oxide inhibited the proliferation of HepG2 cells by blocking cell cycle progression at the G1/S transition [18]. Therefore we performed flow cytometry analysis, with the goal of assessing the effects of the agent on cell cycle. The cells were treated for 72 h with silybin, and thereafter samples were stained with propidium iodide for 1h and examined by flow cytometry. For each sample, the percentage of cells in each phase of the cell cycle (G0/G1-S-G2M) was calculated (Figure 3). We found that silybin increased the cells in G1-phase of ~22% compared to the untreated control and decreased of 47% the cells in S-phase. The block of cell cycle in G1 phase occurred in parallel with a reduction of the expression of cyclin A and cyclin-dependent kinase inhibitor p21 (Figure 3). Previous reports showed that p21 may induce or inhibit apoptosis depending on the drug and the system used [19]. On this basis, we reported the analysis of apoptosis of the HepG2 incubated with silybin carried out by double-labeling with propidium iodide and annexin V. In particular, after 72 h silybin was able to induce apoptosis in a higher number of cells (60%) when compared to untreated cells (Figure 4).

### 2.2. Effects of Silybin on Lipid Metabolism

Many anticancer agents elevate endogenous levels of long-chain ceramides through sphingomyelin hydrolysis or the de novo pathway [20]. In Table 1, we show the analysis and the identification, carried out by mass spectrometry MALDI TOF MS, of lipids extracted from the culture media of HepG2 cells treated with or without silybin for 72 h. In both the spectra were present different ceramides with *m*/*z* range between 500 and 700, namely, sphingomyelin (*m*/*z* = 703.9 ± 0.2) and PI-Cer20:0/16:0 (*m*/*z* = 810.69 ± 0.2). Mass peaks corresponding to ceramides (*m*/*z* = 599.92 ± 1), ceramide phosphorylethanolamines (*m*/*z* = 659.84 and 675.76) and GlcCer (*m*/*z* = 760.98) were present only in the cells treated with silybin.

### 2.3. Profiling of miRNAs Secreted into the Medium Following Treatment with Silybin

In our previous work, we demonstrated that sorafenib altered the levels of ceramides and, consequently, increased the secretion in vitro of miR423-5p, which is involved in autophagy in HepG2 cells [16,21]. On this basis, we analyzed the expression profiling of 84 miRNAs secreted into the culture medium of HepG2 before and after treatment with silybin. Following the acquisition of the data, a complex statistical analysis between the two groups of samples was performed that allowed for the identification of a panel of miRNAs differentially expressed in cells treated with silybin as compared to untreated cells. In particular, after treatment with silybin, miR223-3p and miR16-5p were upregulated, while miR-92-3p was downregulated (*p* < 0.05) (Figure 5).

### 2.4. Silybin Increases the Citotoxic Effect of Sorafenib by Downregulating miR-92a

Based on the proapoptotic capacity of silybin, we combined silybin with sorafenib in order to strengthen its antitumor effects. The combination was highly synergistic when sorafenib and silybin are used in a ratio of IC25:IC75, respectively, with a CI50 (combination index calculated for the 50% of cells survived) equal to 0.4 (Table 2). All subsequent experiments were conducted on HepG2 cells treated with 1.25 μM of sorafenib and 51 μM of silybin in combination, concentrations at which the synergistic effect is more significant with a *p* < 0.005.

To clarify the mechanisms at the basis of synergism, we evaluated the effects of the silybin and sorafenib, alone and in combination, on AKT expression and activity after 72 h of treatment (Figure 6). In particular, sorafenib weakly decreased AKT expression and activity, whereas treatment with silybin induced greater AKT activity inhibition. The combination did not potentiate the effect induced by silybin on the activation of AKT. On the other hand, in HepG2 cells transfected with mutated PTEN, silybin was no longer able to inhibit the activation of AKT while the combination inhibited AKT activation also in absence of PTEN. Sorafenib alone or in combination did not induce any significant effect on the expression of AKT also after transfection of PTEN.

On the basis that miR-92a was identified to target the phosphatase and tensin homolog (PTEN)/AKT pathway [22], we validated miR-92-aexpression in the medium of cells treated with silybin, sorafenib alone, and in combination. Results are reported in Table 3. In details, miR-92-3p down regulation in medium of silybin-treated cells was confirmed (−2.55 fold change) and significantly increased in the medium of the Sil/Sor combination treated HepG-2 cells (−5.55 fold change) compared to the untreated control cells, while the miRNA was not detected in medium of sorafenib treated cells (Table 3).

## 3. Discussion

The fastest growing of HCC incidence, could be at least partly attributable to the rising prevalence of non-alcoholic fatty liver disease. Recently, several publications reported that natural molecules are able to induce cellular differentiation, drastically reducing both proliferation and invasiveness of tumor cells, without having toxic effects on normal cells [23,24]. The beneficial effects of these natural compounds have been attributed, in part, to the presence of numerous polyphenolic substances with antioxidant properties and free radical scavenger. Silybin is the main active ingredient of silymarin, the flavonolignan mixture extracted from the seeds of milk thistle (*Silybum marianum*). Both in vitro and in vivo studies suggest that silybin has hepatoprotective properties against toxins and anticancer properties in human cells of prostate adenocarcinoma, breast cancer (estrogen-dependent and -independent), cervix carcinoma, colon cancer, and lung cancer of different histological type. Silybin has very low toxicity on normal cells and has clearly demonstrated a dose-dependent inhibition of tumor cell growth, angiogenesis, and metastatic invasion [25,26,27]. The main mechanism of action is due to the modulation of the expression of cyclins, CDK, and CDK inhibitors. Recently, it was also demonstrated that silybin reduces cell proliferation caused by UVB rays, and also inhibits microvessel density, inflammation, and angiogenic reactions. Moreover, through interaction with the genome and the modulation of expression of genes, such as MMP-2, u-PA, ERK1/2, AP-1, and NF-kB, the plant complex plays a very important antimetastatic role [28,29,30,31]. Its relative lack of toxicity on normal hepatic cells and its dose-dependent effects on cell growth regulation broaden silybin’s potential for therapeutic use and led us to investigate the effects of silybin on hepatocarcinoma and its ability to enhance the antitumor activity of sorafenib.

In the present study, we have shown that silybin, alone or in combination with sorafenib, induced a cooperative antitumor effect on the HCC cell line HepG-2. In details, the viability assay showed that silybin, after 72 h, had an IC50 of 68 μM. To understand the molecular mechanisms at the basis of cell growth inhibition we evaluated the effects of the compound on cell cycle and apoptosis. Flow cytometric analysis showed that silybin was able to induce a cell cycle block in phase G1. The cycle block in G1 phase was associated with a decreased expression of cyclin A. Moreover, we demonstrated that the growth inhibition induced by silybin was due to a significant increase of apoptosis. Silybin induced a decrease of lipid peroxidation and increase of both MnSOD activity and extracellular nitric oxide (NO) production. Nitric oxide-donating compounds have been reported as promising therapeutic tools due to the ability of NO to increase chemosensitivity in several cancer models [32,33]. Over the past two decades, the literature has shown that the metabolism of sphingolipids plays a crucial role in the cellular response and in human diseases. In particular, the ceramides, precursors of more complex sphingolipids, have been recognized to act as second messengers, which play an important role in signal transduction and cellular regulation. It was reported that the same ceramides or their derivatives can act as second messengers, mediating antitumor signals in response to various stress stimuli [34] (ceramides are also called “messengers of cell death”). Extensive studies over the past 10 years have shown that these molecules possess pleiotropic different intracellular targets which can mediate their effects on both differentiation and cell death. In particular, silybin caused a significant increase of the synthesis of ceramides that may act as second messengers in several processes. Recent studies have shown that the increase of cellular ceramides may trigger mi-RNA secretion in RNase-resistant lipid vesicles and secretory miRNAs can induce silencing effects to the recipient cells. miRNA-enriched exosome release is strongly increased by the activation of the ceramide pathway and decreased by its inhibition. In addition, ceramides have been found to modulate tumor-related miRs inside cancer cells and in released exosomes [21]. On this basis, we analyzed the expression profiling of miRNAs secreted into the culture medium of HepG2 before and after treatment with silybin. In particular, following treatment with silybin, miR223-3p and miR16-5p are upregulated, while miR-92a-3p is downregulated (*p* <0.05). In our system, silybin induced apoptosis, upregulating secretory tumor suppressive miRs and downregulating secretory oncomiRs by secondary lipid metabolite production. In particular, aberrant expression of microRNA-92a (miR-92a), belonging to the cluster of miR-17-92, has been found in HCC, colorectal cancer, breast cancer, lung cancer, and cervical cancer. In HCC, high level of miR-92a was correlated with poor prognosis. Functionally, miR-92a was involved in cancer migration and invasiveness and was identified to target phosphatase and tensin homolog (PTEN)/AKT pathway [22,35]. On these basis, we hypothesize that silybin induces apoptosis through the synthesis of ceramides that may then act as second messengers and modulate the expression of specific miRNAs that likely target (PTEN)/AKT pathway and are involved in the pathogenesis of HCC. The HCC in advanced stages is a tumor that is mortally aggressive, and does not respond to drug therapy and radiotherapy. This emphasizes the great need to develop new strategies for the prevention and treatment of this disease. In order to reduce the morbidity and mortality of cancer, early diagnosis and the development of new systemic therapy for advanced disease, including drugs, gene and immune therapy, as well as primary prevention for HCC are most important. The main, if not the only medical therapy for patients with advanced HCC is sorafenib. The latter is an inhibitor of multiple kinases and is the first molecular targeted therapy for HCC that has proven effective in a large clinical trials of patients with advanced tumors, prolonging their survival and time to radiologic progression [36]. Despite this therapeutic advance in the treatment of HCC, we are still far from effective control of the disease. The failure of the medical therapy of HCC is due, at least in part, to the complex molecular alterations in tissue of HCC and the activation of multiple signal transduction pathways and tumor progression [37,38]. Angiogenesis is one of the characteristics of HCC and is also a target of sorafenib. In particular, the target of sorafenib is the VEGF-R, which is responsible for neovascularization of HCC. However, other growth factors and growth factors receptors, such as epidermal growth factor (EGF) and insulin like growth factors (IGF), are involved in hepatocarcinogenesis [37,38]. Based on these considerations, in order to enhance the antitumor effects of sorafenib and reduce its side effects, we evaluated the synergism between sorafenib and silybin on the inhibition of HCC cell proliferation. The results show that the two drugs given to molar concentrations equal to IC25:IC75 of sorafenib and silybin, respectively, were highly synergistic. Most published reports have reported the ability of silybin to target an array of cellular signaling pathways and molecules, including phosphoinositide 3-kinase/Akt [39,40,41]. To clarify the mechanisms at the basis of synergism, we evaluated the effects of silybin and sorafenib, alone and in combination on AKT expression and activity. In particular, sorafenib weakly decreased AKT expression and activity, whereas the treatment with silybin induced greater AKT activity inhibition. The combination did not potentiate the effect induced by silybin on the activation of AKT. On the other hand, in HepG2 cells transfected with mutated PTEN, silybin was no longer able to inhibit the activation of AKT, while the combination inhibited AKT activation also in absence of PTEN. Sorafenib alone or in combination did not induce any significant effect on the expression of AKT also after transfection of PTEN. Since miR-92a was identified to target phosphatase and tensin homolog (PTEN)/AKT pathway, we validated miR-92-aexpression in medium of cells treated with silybin, sorafenib, and in combination. In details, miR-92-3p down regulation in medium of silybin-treated cells was confirmed (−2.55 fold change) and significantly increased in the medium of the Sil/Sor combination treated Hepg2 cells (−5.55 fold change) compared to the untreated control cells while the miRNA was not detected in medium of sorafenib treated cells. Recent studies have demonstrated that miR-92 was upregulated in HCC compared to paracancerous tissue. Moreover, the authors reported a negative correlation between miR-92 and the tumor suppressor gene PTEN in HCC clinical tissues, by suggesting that PTEN and miR-92 play opposite roles in HCC pathogenesis. miR-92 was identified to target PTEN and induce tumorigenesis by activating PI3K/Akt pathway. In normal conditions, PTEN is a negative regulator of AKT, but in tumors high levels of miR-92 may inhibit PTEN expression and counteract the inhibition of PI3K/Akt pathway. In our system, silybin downregulated miR92a and inhibited AKT activity in a PTEN-dependent manner since when we transfect cells with a dominant negative of PTEN, AKT is no longer inhibited. This finding did not occur with the combination, which was able to inhibit AKT activation also in absence of PTEN, probably due to a major miR92-a downregulation [22,35].

Based on these findings, miR92a could represent a promising target to treat cancer poorly sensitive to pharmacological treatments such as HCC at an advanced stage.

## 4. Materials and Methods

### 4.1. Cell Cultures

The experimental system used is represented by a human HCC cell line (HepG2) provided by the American Type Tissue Culture Collection, Rockville, MD, USA. Such cells have many of the specific metabolic functions of the liver, in particular those related to lipoprotein metabolism. The cells were grown in RPMI 1640 culture medium, to which heat-inactivated 10% fetal serum bovine, 20 mM Hepes, 100 U/mL penicillin, 100 μg/mL streptomycin, and 1% l-glutamine had been added. The cells were exposed to a humidified atmosphere of 95% air/5% CO_2_ at 37 °C.

### 4.2. Thiobarbituric Acid-Reactive Species (TBARS) Levels

Cell lysates were incubated with 0.5 mL of 20% acetic acid, pH 3.5, and 0.5 mL of 0.78% aqueous solution of thiobarbituric acid. Samples were heated at 95 °C for 45 min, and then centrifuged at 4000 r.p.m. for 5 min. TBARS were quantified by a spectrophotometer performing adsorbance readings at 532 nm, and results were expressed as previously described [2]. Each data point was performed in triplicate.

### 4.3. Nitrite Levels

Nitrite was measured by the Griess assay as previously reported [2]. Nitrite levels were quantified by the spectrophotometer reading adsorbance at 550 nm. Each data point was performed in triplicate.

### 4.4. Cell Proliferation Assay

Cell viability tests were performed using MTT assays. MTT [3-(4,5-dimethylthiazol-2-yl)-2,5-diphenyltetrazolium bromide], is a vital dye, which is metabolized by the cell at the level of the mitochondria. Cells were plated at 2.5 × 10^3^ in 96-well plates. After 24 h, cells were exposed to various concentrations of Silybin (0–200 μM) for 24, 48, and 72 h. After the time of incubation with the different treatments, the MTT was added to all the wells in a concentration of 10%. After a minimum incubation of 4 h, the medium/MTT solution was aspirated; later the formazan salts were solubilized with an isopropanol/HCL 0.1 N solution for 20 min in constant agitation. Finally, the absorbance measurement of the present solution was performed respectively in each well of the multiwell, with an ELISA reader Biorad 550 (Bio-Rad Laboratories GmbH, München, Germany). Absorbance readings were performed at 570 nm. Experiments were performed in triplicate.

### 4.5. Cell Cycle and Apoptosis Analysis

Hep-G2 cells were seeded in 6-well plates at the density of 2 × 10^5^ cells/plate. After incubation with Sil, cells were washed in PBS, and incubated with a propidium iodide (PI) solution (50 μg PI in 0.1% sodium citrate, 0.1% NP40, pH 7.4) for 45 min at 4 °C in the dark. Evaluation of apoptosis was performed using iodide staining of propidium iodide (PI) and Annexin-V-FITC by using the Annexin-V-kit (BD Biosciences Pharmingen, Heidelberg, Germany). Briefly, the cells washed in PBS and were incubated with the Annexin-V-FITC in a special buffer (supplied by the manufacturers) for 10 min at room temperature, and then washed and resuspended in the same buffer, as described by the manufacturers. BD Accuri C6 flow cytometer (Becton Dickinson, San Jose, CA, USA) was used to perform FACS analysis. PI fluorescence was collected on FL2 channel while Annexin-V FITC fluorescence on FL1channel. For each sample, at least 20,000 events were acquired in at least 3 different experiments.

### 4.6. Western Blotting

Cells were plated and treated with IC:50 silybin and sorafenib individually or in combination obtained from Calcusyn. After 72 h of treatment, cells were washed and collected in PBS 1X on ice. Pellets obtained after centrifugation a 14000× *g* at 4 °C, were lysed for an hour on ice in a buffer containing HEPES 50 mM, NaCl 150 mM, Glycerol 1%, TRITON 1%, MgCl_2_ 1.5 mM. EGTA 5 mM, 20 nM Na pyrophosphate, 10 mM Na orthovanadate, 25 mM NaF, Aprotinin (5 µL/mL), and PMSF (Phenylmethylsulfonyl fluoridate) 0.5 mM. The samples were centrifuged at 4 °C for 10 min and the supernatant was used for the experiments. Total protein concentration was determined by Biorad assay. Protein lysates, before being loaded onto the SDS-PAGE polyacrylamide gel, were denatured for 5 min at 100 °C in Sample Buffer 1X (Sample Buffer 5X: Tris 10 mg/mL, SDS 30 mg/mL, β-mercaptoethanol 0.15 mL, glycerol 0.3 mL, bromophenol blue). After electrophoretic separation, the proteins were transferred by Western blotting on nitrocellulose filter. The filter was washed twice with 0.05% Tween20/TBS (200 mM Tris-HCl pH 7.5, 1.5 M NaCl) and incubated for 1 h at 37 °C in blocking buffer (T-TBS/Milk at 5%). After the incubation, the filter was exposed to a solution of T-TBS/Milk with the different antibodies properly diluted.

At the end the nitrocellulose filter was incubated with enhanced chemoluminescence detection reagents (SuperSignal West Pico, Pierce, Cheshire, UK) and developed by Chemidoc software (BioRad laboratories, Hercules, CA, USA).

### 4.7. RNA Purification

Total RNA, including small RNAs, was extracted from cell medium using miRNeasy Mini kit (Qiagen) according to the manufacturer’s protocol.

### 4.8. miRNA Profiling and Real-Time PCR Analyses

During the first step, total RNA is reverse transcribed using predefined RT primers that are specific for only the mature miRNA species. In the second step, each of the RT pools containing cDNA template are diluted, mixed together with TaqMan Universal PCR Master Mix, and loaded on the TaqMan Array. The card is briefly centrifuged to distribute samples to multiple wells on the array, and then sealed. Real-time quantitative PCR was performed on a ViiA7™ Real-time PCR system (Applied Biosystems, Darmstadt, Germany). Relative expression of the transcripts was measured by using ViiA7™Real-Time PCR software (Applied Biosystems, Darmstadt, Germany).

### 4.9. Extraction of Lipid and MALDI-TOF MS Analysis

Total lipids were extracted through chloroform extraction: using methanol according to Stiuso et al. [2,42]. The membrane was homogenated with chloroform/methanol (2:1 *v*/*v*) up to a final volume twenty times the volume of sample (1 g in 20 mL of solvent mix). After dispersion, all the mixture was stirred for 15 to 20 min on an orbital shaker at room temperature. The homogenate was first filtered then centrifuged at 12,000× *g* for 10 min to recover the liquid phase. The liquid phase is washed with 0.2 volumes (4 mL per 20 mL) of saline solution of NaCl 0.9%. After vortexing for a few seconds, the mixture is centrifuged a low speed (2000 rpm) to separate the two phases. The upper phase was removed and, after centrifugation, the lower chloroform phase containing lipids was used for structural analysis using MALDI-TOF mass spectrometry. MALDI-TOF mass spectrometry analysis of cellular media was conducted using a mass spectrometer model Voyager-DE PRO BioSpectrometry Workstation (Applied Biosistems, Lancashire, UK) equipped with a N_2_ laser (337 nm, 3 ns pulse width). The analyte solution (1 μL) was loaded onto a special one stainless steel plate (target) mixing it with 1 μL of matrix. It was used as matrix 10 mg of 2,5-dihydroxybenzoic acid in 1 mL of chloroform/methanol, 2:1 *v*/*v*. The analytes were cocrystallized with the matrix allowing the evaporation to take place solvent in the air. The spectra were acquired in linear mode and reflector-mode by appropriately varying the range of *m*/*z* analyzed. To check repeatability, spectra were acquired in triplicate at least.

### 4.10. Drug Combination Studies

Cells were seeded in 96-well plates at the density of 2.5 × 10^3^ cells/well. After 24 h incubation at 37 °C the cells were treated with different concentrations of Silybin and Sorafenib and their combination. Synergism was evaluated by the dedicated software Calcusyn as previously described [43].

### 4.11. Statistical Analysis

All data were expressed as mean ± SD. Statistical analysis was performed by analysis of variance (ANOVA) with Neumann–Keul’s multiple comparison test or Kolmogorov–Smirnov test where appropriate.

## 5. Conclusions

In conclusion, these data suggest that silybin through the synthesis of ceramides may to activate the processes of programmed death by inducing the secretion of specific miRNAs that likely target (PTEN)/AKT pathway. These findings offer new therapies to treat cancer that is poorly sensitive to pharmacological treatments, such as HCC at an advanced stage. The aim of this study was to investigate in depth the molecular mechanisms of silybin antitumor efficacy in a noninvasive and faster way with respect to all the in vivo procedures.

## Figures and Tables

**Figure 1 ijms-20-02190-f001:**
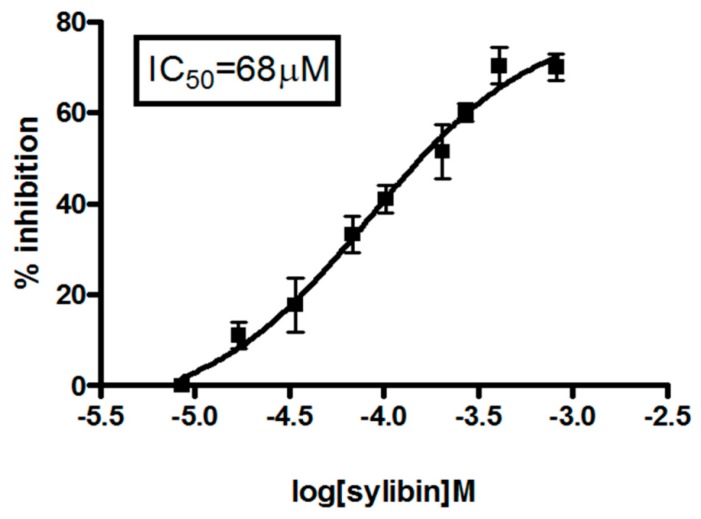
Dose-response effects of silybin on HCC cells. HepG2 cells were seeded and treated with different concentrations of Silybin (0–200 mM) for 72 h, and thereafter cell viability was assessed as described in “Materials and Methods”. The experiment was repeated three times and the results are the mean of the different data. Bars represent standard deviations.

**Figure 2 ijms-20-02190-f002:**
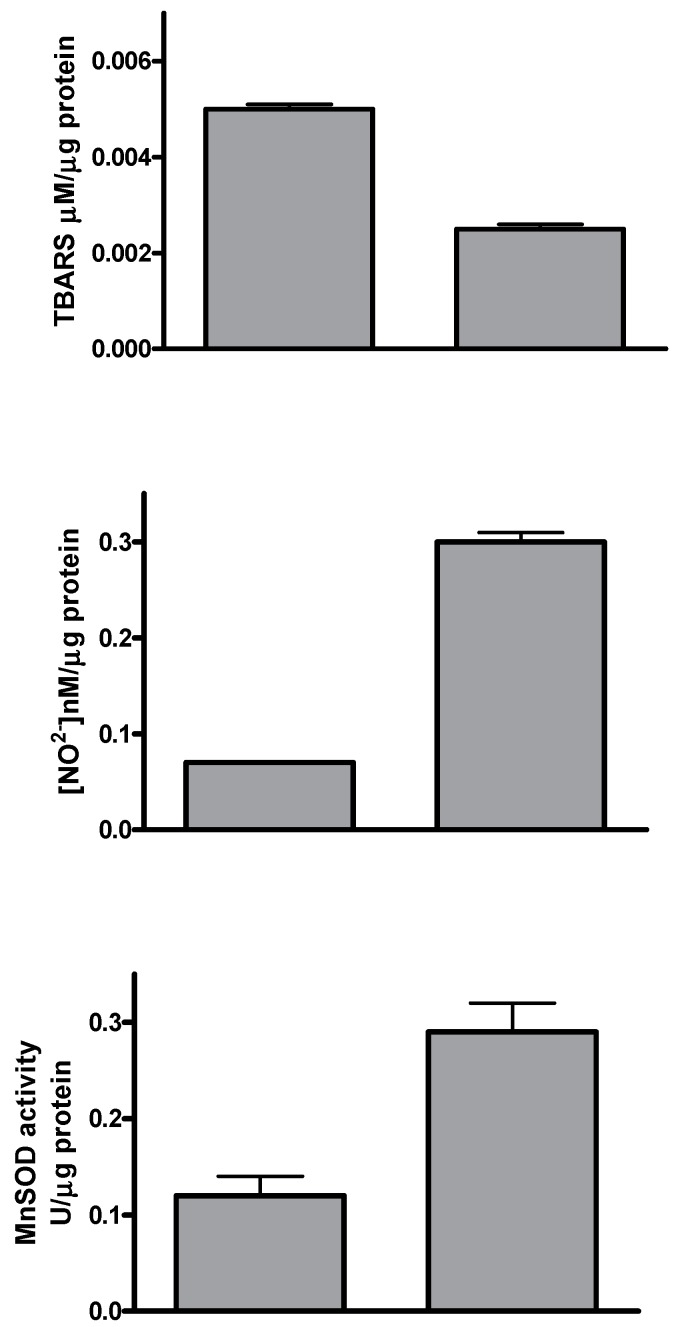
Antioxidant activity of silybin. HepG2 cells were seeded and treated with IC50 of Silybin for 72 h, and thereafter thiobarbituric acid-reactive species (TBARS), NO production, and MnSOD activity were assessed as described in “Materials and Methods”. The experiment was repeated three times and the results are the mean of the different data. Bars represent standard deviations.

**Figure 3 ijms-20-02190-f003:**
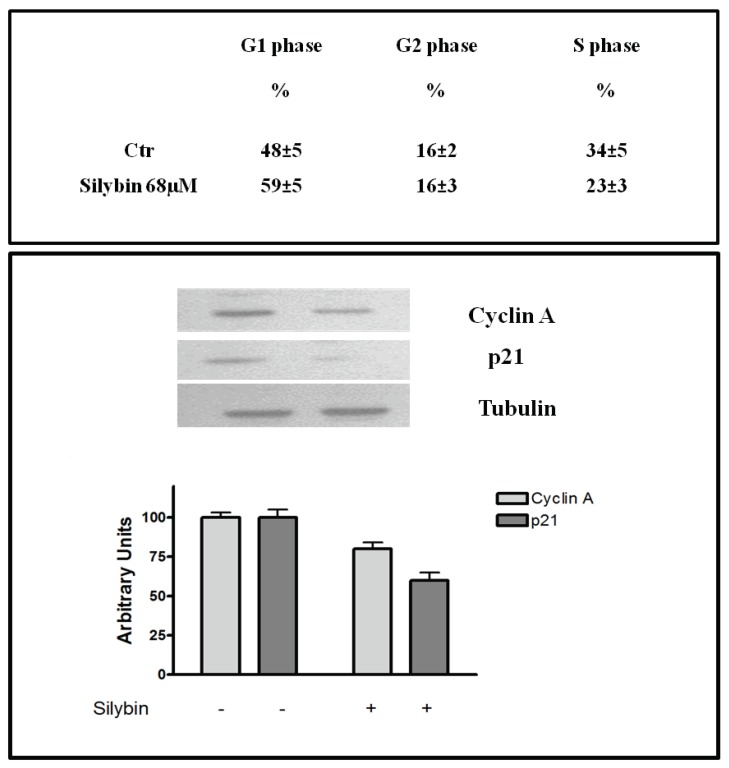
Effects of silybin on cell cycle. (**up**) Cell cycle distribution after 72 h of treatment with silybin IC50 (68 μM) in HepG2 cells, evaluated by FACS, as described in Materials and Methods. Data are expressed as mean ± SD of the percentage of cells in the different phases of the cell cycle, as compared with untreated control cells. (**down**) Effects of silybin on p21 and cyclin A expression in HepG2 cells. γ-tubulin was used as loading control. The intensity of each band was expressed as % arbitrary units compared to that of the untreated cells. Error bars showed standard deviation from the mean in at least three independent experiments.

**Figure 4 ijms-20-02190-f004:**
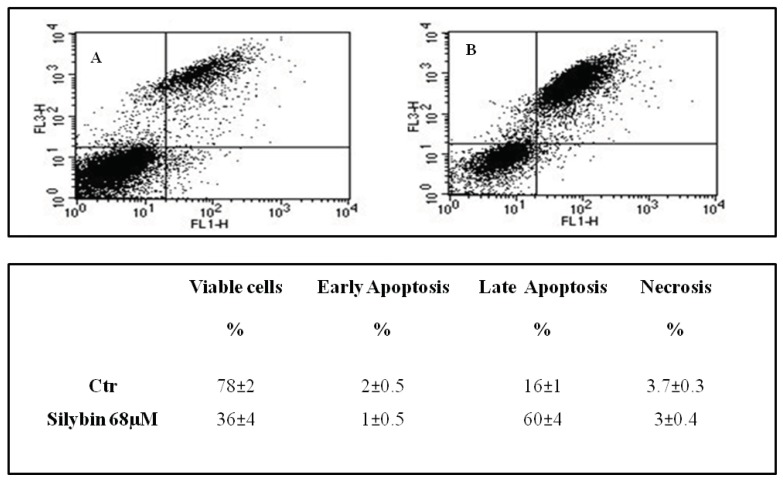
Effects of silybin on apoptosis. Apoptosis after 72 h from treatment with silybin IC50 (68 μM) in HepG2 cells was evaluated by FACS as described in Materials and Methods. (**A**) Untreated cells (Ctr); (**B**) silybin-treated cells. Data are expressed as mean ± SD of the percentage of cells in the different phases of the cell cycle, as compared with untreated control cells.

**Figure 5 ijms-20-02190-f005:**
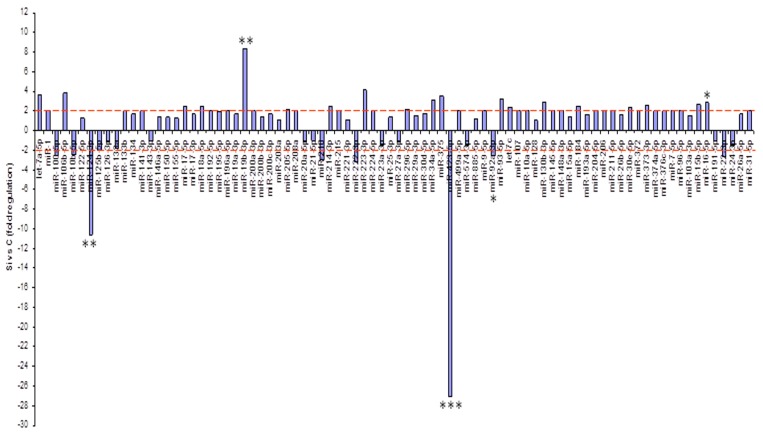
miRNA profiling. Fold regulation of miRNA levels detectable in the medium of silybin-treated cells versus untreated cells. *p*-values of the Student’s *t*-test were * *p* < 0.05, ** *p* < 0.01, or *** *p* < 0.001.

**Figure 6 ijms-20-02190-f006:**
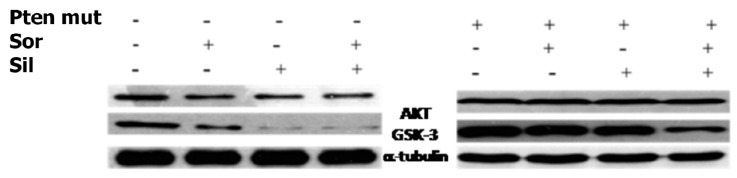
Silybin increases the cytotoxic effect of sorafenib. Effects of silybin (sil), sorafenib (sor) on AKT, GSK3 expression, in HepG2 cells transfected or not, with mutated PTEN (PTEN mut) was evaluated by Western blotting. γ-tubulin was used as loading control; standard deviation from the mean in at least three independent experiments.

**Table 1 ijms-20-02190-t001:** MALDI-TOF mass spectra of cellular lipids.

*m*/*z*	Assignment	Ctr	Sil
545.56 ± 1	Cer(t20:0/26:0)	+	+
551.49 ± 1	Cer(m18:1(4E)/18:0)	+	−
599.92 ± 1	Cer(t18:0/18:0(2OH))	−	+
567.63 ± 1	Cer(d18:0/18:0)	+	+
637.8 ± 1	Cer(d14:1(4E)/26:0(2OH))	−	+
659.84 ± 1	PE-Cer(d14:2(4E,6E)/20:0)	−	+
675.76 ± 1	PE-Cer(d16:1(4E)/19:0)	−	+
703.9 ± C16	Sphingomyelin	+	+
725.84 ± 1	Cer(t20:0/26:0)	+	+
732.77 ± 1	GlcCer(t18:1(8Z)/16:0(2OH[S]))	−	−
760.98 ± 1	GlcCer(t18:1(8Z)/18:0	−	+
782.76 ± 1	PI-Cer(d20:0/14:0)	+	+
788.77 ± 1	C22 Sphingomyelin	−	−

Ctr: untreated cells; sil: silybin-treated cells.

**Table 2 ijms-20-02190-t002:** Synergism between silybin and sorafenib.

Drug	IC50 μM (Drug Alone)	IC50 μM (Drug Combination)	CI50	Interpretation
Sorafenib	5±0.02	1.25±0.003	0.4	Strong synergism
Silybin	68±0.03	51±0.01		

**Table 3 ijms-20-02190-t003:** Fold regulation of miRNA levels detectable in the medium of silybin (sil), sorafenib (sor), and silybin/sorafenib-treated cells versus untreated cells (ctr). * highly significant.

	hsa-miR-92a-3p	
	Fold Regulation	*P* Value
Sil vs. ctr	−2.55	0.036 *
Sor vs. ctr	n.d	n.d
Sil/Sor vs. ctr	−5.55	0.002 *

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
