# Peer review of "Silybin-Induced Apoptosis Occurs in Parallel to the Increase of Ceramides Synthesis and miRNAs Secretion in Human Hepatocarcinoma Cells"

_ijms, 2019, doi:10.3390/ijms20092190_

Reviewer 1 Report

Silybin–induced Apoptosis Occurs in Parallel to the Increase of Ceramides Synthesis and miRNAs Secretion in Human Hepatocarcinoma Cells

By the authors Zappavigna S., Vanacore D., Lama S., Potenza N., Russo A., Ferranti P., Dallio M., Federico A., Loguercio C., Sperlongano P., Caraglia M. and Stiuso P.

In this manuscript by Zappavigna et al. the authors aimed at identifying sylibin induced apoptosis modulating secretory miRNA by secondary lipid metabolites production on HepG2 cells. In fact, treatment with silybin, induced miR223-3p and miR16- upregulation while miR-92-3p was down regulated. Thus, the authors conclude a possible clinical implication suggesting thast miRNAs may represent promising new targets for anti-cancer therapy. The study by Zappavigna et al. is interesting and generally well written; however, the presented data need to be regarded as preliminary and hypothesis-generating; warranting cautious interpretation of the present dataset. There are some specific concerns:

1)   Was there are control group to test the effect of silybin on normal hepatic tissue ?

2)   Why did the authors decide to use just one cell line to test their hypothesis instead of validating their findings in a second cell line setting ?

3)   Conclusions are very general and not specific. Would recommend concluding the data of the study and not in general that miRNAs might be an important tool in HCC

4)   This study would profit from further in-vivo experiments

5)   Too much discussion in the “results part”

6)   The authors should actually cite the SHARP trial (PMID: 18650514, line 241-245)

Author Response

Reviewer  1

In this manuscript by Zappavigna et al. the authors aimed at identifying sylibin induced apoptosis modulating secretory miRNA by secondary lipid metabolites production on HepG2 cells. In fact, treatment with silybin, induced miR223-3p and miR16- upregulation while miR-92-3p was down regulated. Thus, the authors conclude a possible clinical implication suggesting thast miRNAs may represent promising new targets for anti-cancer therapy. The study by Zappavigna et al. is interesting and generally well written; however, the presented data need to be regarded as preliminary and hypothesis-generating; warranting cautious interpretation of the present dataset. There are some specific concerns:

Question: Was there are control group to test the effect of silybin on normal hepatic tissue ?

Answer:The effects of silybin on normal hepatic tissue have been widely investigated in previous reports. Silybin have been found clinically effective in the treatment of liver diseases on the basis of its cytoprotective actions. These actions include reduction of cell membrane permeability (Munter et al., 1986), scavenging of reactive oxygen species (Mira et al., 1994), inhibition of leukotriene formation (Dehmlow et al., 1996) and suppression of DNA-binding activity of NF-κB and its consequent effects on gene expression (Saliou et al., 1998). However, the growth-regulating actions of silibinin are proposed to be highly dose-dependent, with low doses promoting and high doses suppressing growth. Findings on normal hepatic cells also support the idea that silybin obstructed progression of the cell cycle to G2/M while preserving viability and function as evidenced by increased DNA synthesis and ERK signaling. Its relative lack of toxicity and adverse events broadens silibinin’s potential for therapeutic use and led us to investigate the effects of silybin on hepatocarcinoma and its ability to enhance the antitumor activity of sorafenib

Question: Why did the authors decide to use just one cell line to test their hypothesis instead of validating their findings in a second cell line setting ?

Answer: We used one cell line because we found a significant synergism between silybin  and sorafenib only in HepG2 compared to Huh7. Our data on Huh7 were already published in our previous work. (Stiuso et al. MTNA 2015)

Question: Conclusions are very general and not specific. Would recommend concluding the data of the study and not in general that miRNAs might be an important tool in HCC

Answer: As correctly suggested by the reviewer, we have rewritten conclusions by better describing  our final findings.

Question: This study would profit from further in-vivo experiments

Answer:Several in vivo studies already show the efficacy of Silybin in reducing HCC xenograft growth through the inhibition of cell proliferation, cell cycle progression and PTEN/P-Akt and ERK signaling, inducing cell apoptosis, and increasing histone acetylation and SOD-1 expression.

The aim of this study was to study in depth the molecular mechanisms of silybin antitumor efficacy in a non invasive and faster way with respect to all the in vivo procedures. In fact, studying the effect of sylibin on miRNA profiling requires the use of a large number of samples to perform both screening and validation phase, according to this, to use in vitro models can lead to several benefits including reduction of pre-clinical studies on genetically engineered mice, costs and possibly faster implementation of clinical trials.

Question: Too much discussion in the “results part”

Answer: As correctly suggested by the reviewer, we have deleted the discussion part in the “results” section.

4.    Question:  The authors should actually cite the SHARP trial (PMID: 18650514, line 241-245)

Answer: According to the reviewer, we have correctly cited the required reference

Reviewer 2 Report

The authors domonstrate that silybin a flavonolignan extracted from Silybum marianum (milk thistle) has the hepato-protective, anti-oxidant and anti-inflammatory activity, with a strong increase of ceramides involved in the modulation of miRNA secretion. After treatment with silybin, miR223-3p and miR16-5p were upregulated while miR-92-3p was down regulated (p<0.05). They suggest that sylibin induced apoptosis modulating secretory miRNA by secondary lipid metabolites production on HepG2 cells.

However, they did not provide any evidence or clue why, even though both miR223-3p and miR16-5p were upregulated with the strong increase of ceramides involved in the modulation of miRNA secretion, miR-92-3p was down regulated.

The authors should provide the answer to this.

Author Response

The authors domonstrate that silybin a flavonolignan extracted from Silybum marianum (milk thistle) has the hepato-protective, anti-oxidant and anti-inflammatory activity, with a strong increase of ceramides involved in the modulation of miRNA secretion. After treatment with silybin, miR223-3p and miR16-5p were upregulated while miR-92-3p was down regulated (p<0.05). They suggest that sylibin induced apoptosis modulating secretory miRNA by secondary lipid metabolites production on HepG2 cells.

      However, they did not provide any evidence or clue why, even though both miR223-3p and miR16-5p were upregulated with the strong increase of ceramides involved in the modulation of miRNA secretion, miR-92-3p was down regulated.

Question: The authors should provide the answer to this.

Answer: We have better discuss our statement as follows:”Ceramide increase plays a key role in miRNA secretion and secretory miRNAs can induce silencing effects to the recipient cells. miRNA-enriched exosome release is strongly increased by the activation of the ceramide pathway and decreased by its inhibition. In addition, ceramides have been found modulate tumor-related miRs inside cancer cells and in released exosomes. In our system sylibin induced apoptosis upregulating secretory tumor suppressive miRs and downregulating secretory oncomiRs by secondary lipid metabolites production.” In particular, we show that sylibin inhibits pi3k /akt pathway probably modulating miR92a expression that has PTEN as target.

Round  2

Reviewer 1 Report

This manuscript has been significantly improved and is now up to standard for publication in IJMS

Reviewer 2 Report

I agree to accept this manuscript at this time.